# Inference of field reversed configuration topology and dynamics during Alfvenic transients

J.A. Romero [1], S.A. Dettrick[1], E. Granstedt[1], T. Roche[1] & Y. Mok[1]

Active control of field reversed configuration (FRC) devices requires a method to determine the flux surface geometry and dynamic properties of the plasma during both transient and steady-state conditions. The current tomography (CT) method uses Bayesian inference to determine the plasma current density distribution using both the information from magnetic measurements and a physics model in the prior. Here we show that, from the inferred current sources, the FRC topology and its axial stability properties are readily obtained. When Gaussian process priors are used and the forward model is linear, the CT solution involves non-iterative matrix operations and is then ideally suited for deterministic real-time applications. Because no equilibrium assumptions are used in this case, inference of plasma topology and dynamics up to Alfvenic frequencies then becomes possible. Inference results for the C-2U device exhibit self-consistency of motions and forces during Alfvenic transients, as well as good agreement with plasma imaging diagnostics.

[1] TAE Technologies Inc., PO Box 7010 Rancho Santa Margarita, CA 92688, USA. Correspondence and requests for materials should be addressed to J.A.R. (email: jromero@tae.com)

A field reversed configuration (FRC) has no externally imposed toroidal field, belonging to the category of compact tori[1, 2]. The poloidal field in an FRC has one component arising from magnets arranged on a common linear axis and another component generated by a toroidal plasma current flowing in opposite direction to the magnet currents. Under transient conditions, an additional magnetic field component arises from toroidal currents flowing in the vessel (the flux conserver (FC) current), which are induced by changes in plasma current distribution and/or transients in the external magnet currents. When the plasma current is strong enough to reverse the externally imposed magnetic field, a closed field structure topologically similar to a torus is formed (Fig. 1). Closed field lines in an FRC enclose a point of maximum kinetic pressure and null magnetic field called the o-point. The separatrix is the flux surface with null poloidal flux and separates the internal closed field region from the open field line scrape-off layer (SOL) region.

The C-2U device, built and operated by Tri Alpha Energy (TAE), is the first device to demonstrate that FRCs can be sustained in near steady state using neutral beam injection[3, 4]. TAE's C-2U device relied largely on FC effects to stabilize plasma displacements, so the discharge lifetime was of the same order as the time constant of the vessel. TAE's C-2W device, presently in its initial operational phase, will extend the discharge duration over this limit, so plasma control will become necessary to stabilize the separatrix shape and position[5]. A method to determine the magnetic field structure and related control variables in real time (with sampling frequency in the range 10–100 kHz) is then required. Some first order approximations for FRC geometry parameters are available from the excluded flux radius. However, these cannot distinguish an FRC from a high beta mirror[6], so they are not particularly useful if the FRC state itself is uncertain.

Determination of the magnetic field structure in an FRC is a challenging problem. While magnetic field structure inside the plasma can be measured by inserting probes inside the plasma[7], this cannot be done in high temperature plasmas without severely disrupting the plasma confinement. In FRC plasmas, the magnetic field structure must be determined indirectly from external magnetic probes[8], laser polarimetry systems[9], etc.

Determination of the internal current sources from external measurements is termed the inverse problem. The technique used to determine the current sources from the sensor data is the inference technique. When Lorentz forces are balanced by plasma pressure, there is no net acceleration of the plasma, and the plasma is said to be in equilibrium. The determination of the magnetic structure corresponding to plasma in equilibrium is referred to in the literature as 'equilibrium reconstruction'.

This work departs from the standard equilibrium reconstruction approach[10] and use instead the current tomography (CT) method[11, 12], a well-validated alternative already studied in connection with real-time control of tokamaks[13]. The CT method uses Bayesian inference[14] of Gaussian processes (GPs)[15] to solve the inverse problem. The GP modelling used by the CT method can be tailored to a multiplicity of related tomography problems[16], in particular to the specifics of the FRC magnetics. There are several advantages of the CT method that make it ideal for plasma control. First of all, when the relationship between current sources and sensors is linear (such as is the case with magnetic probes), and the physics assumptions can be reduced to linear relationships among current sources and measurements, the solution depends on the sensor data through non-iterative matrix operations and, for this reason, is deterministic and suitable for real time. A version of the algorithm for C-2W device has already been implemented in a field-programmable gate array and verified to run under 10 μs. Second, as no equilibrium restrictions are necessarily required, the CT method can infer Alfvenic oscillations from magnetic sensor data. Fast transients can then be resolved accurately and with very low latency, both factors known to have an impact on control systems performance. Third, the CT method is able to fuse information from multiple sensor data sets and boundary conditions using a unified inference approach. This allows straightforward scalability should other magnetic sensors become available at a later stage. Sensors based on Polarimetry[17] and Hanle effect[18], for instance, are both planned for TAE's C-2W device. Finally, the CT method provides uncertainty measures on all inferred outputs. This is interesting information on its own, but it has also an interest for advanced control applications, since the uncertainty information can be factored in as part of a robust control scheme[19].

## Results

**Inference of Alfvenic transients in FRC.** In the C-2U device[3], two individual toroidal current rings are produced inductively (θ-pinch technique) in two opposing quartz formation sections placed at both ends of a stainless steel vacuum vessel (Fig. 2). These are produced simultaneously using pulsed power, fast magnetic field transients, and then accelerated out of their respective formation sections at supersonic speeds $v_z \sim 300$ km/s. Collisions of both FRCs take place inside the confinement vessel near or at the mid-plane $z = 0$.

The merging process occurs during the first few 10s of μs of the discharge, transforming the kinetic energy of the two initial compact tori into thermal energy of a single, static FRC[20]. Neutral beam heating is then applied to this initial FRC to provide the necessary heating and current drive to sustain the discharge against thermal and resistive flux losses.

When the accelerations in both formation sections are slightly different with respect to each other, FRCs do not collide exactly in the middle of the confinement section, leading to a merged FRC with a residual velocity. The resulting FRC is bounced back and forth in the axially stabilizing external mirror field until its position is stabilized around the machine mid-plane. Analysis of these oscillations provides a way to test the compliance of the inferred forces and accelerations with Newton's second law, using estimations of the plasma mass obtained by other diagnostics, as it will be shown.

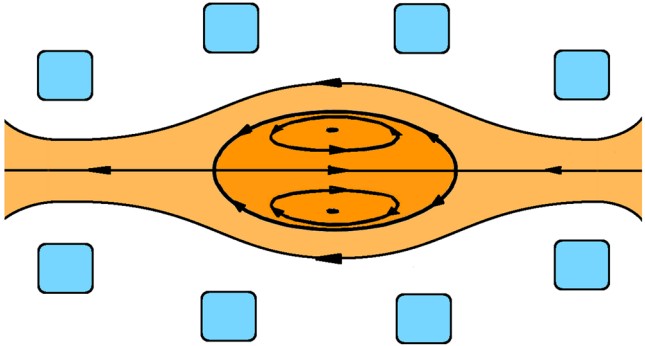

**Fig. 1** Schematics of the magnetic field topology in a field reversed configuration. Plasma (orange) is contained using a set of axially symmetric magnets (blue). When plasma current is strong enough to reverse the externally imposed magnetic field, a closed field line structure is formed. Closed field lines circle around the so called o-point, where the magnetic field is null. The longest closed magnetic field line enclosing the o-point has null poloidal flux and separates the internal closed field region from the open field line scrape-off layer (SOL) region

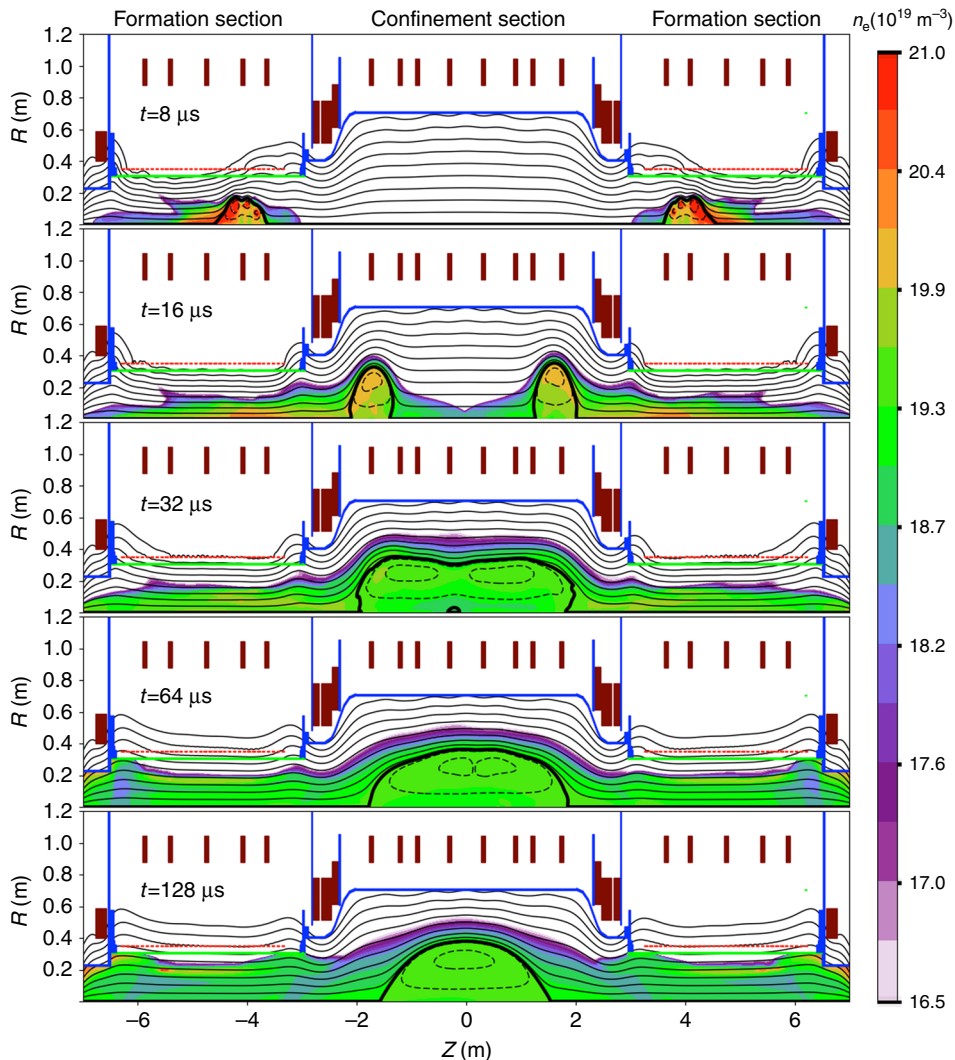

**Fig. 2** MHD simulation of a typical FRC process in the C-2U device. Two FRCs are created in two quartz formation sections. They are accelerated towards each other to collide and merge into a single FRC inside a stainless steel confinement chamber. The collision transforms the kinetic energy of both moving FRCs into thermal energy of a single, static FRC. Magnetic field topology in the SOL (solid lines), closed field region (broken lines) and separatrix (thick solid line) are shown along with colour-coded electron density. DC Magnets (brown blocks), flux conserver structures (blue), fast switching magnets (dotted red line) and quartz tube boundaries (green horizontal lines) are also shown for completeness

A plasma discharge exhibiting Alfvenic plasma oscillations around the midplane is chosen for the study, as illustrated in Fig. 3. Contours of poloidal flux and forward prediction of the actual magnetic measurements are shown every 10 μs. The frequency of the oscillations is about ~ 20 kHz.

The axial position of the o-point is shown in Fig. 4 with high time resolution (every 10 μs), along with other geometric descriptors and plasma variables related to those. The o-point position is strongly correlated with the vessel current imbalance $I_V^\zeta$, defined as:

$$I_V^\zeta = I_V^{z>0} - I_V^{z<0} \tag{1}$$

where $I_V^{z>0}$ is the net toroidal current flowing in one half of the vessel with $z > 0$, and $I_V^{z<0}$ is the net toroidal current flowing the other half of the vessel with $z < 0$. For a static plasma, the vessel current imbalance is zero. As plasma moves back and forth, mid-plane antisymmetric current components are induced in the vacuum vessel, which eventually dissipate ohmically. These are in the direction to oppose and slow down the plasma movements.

As a result, a strong correlation between the o-point axial position and the vessel current imbalance exists, as shown.

The separatrix radius is found to be proportional to the o-point radius, which is in agreement with Eq. (7). The trapped flux $\psi_0$ also matches approximately the approximation (9) for an elongated FRC. These approximations are not used in the inference process but as a check for consistency of the final results with these limiting cases.

The total number of deuterons in the plasma is estimated from line integrated density measurements integrated over the excluded flux radius. Plasma mass is estimated to be $m_p = 1.3 \times 10^{-7}$ kg from the deuteron mass times the deuteron inventory. The acceleration $\ddot{z}$ of the o-point can be determined from its position $z$ (see Fig. 4). The net Lorentz force $F_z$ exerted over the whole plasma current distribution can be determined from the inferred current distribution and derived magnetic field. It turns out the product of the plasma mass and acceleration $\ddot{z}$ is consistent with the inferred electromagnetic force

$$F_z = m_p \ddot{z} \tag{2}$$

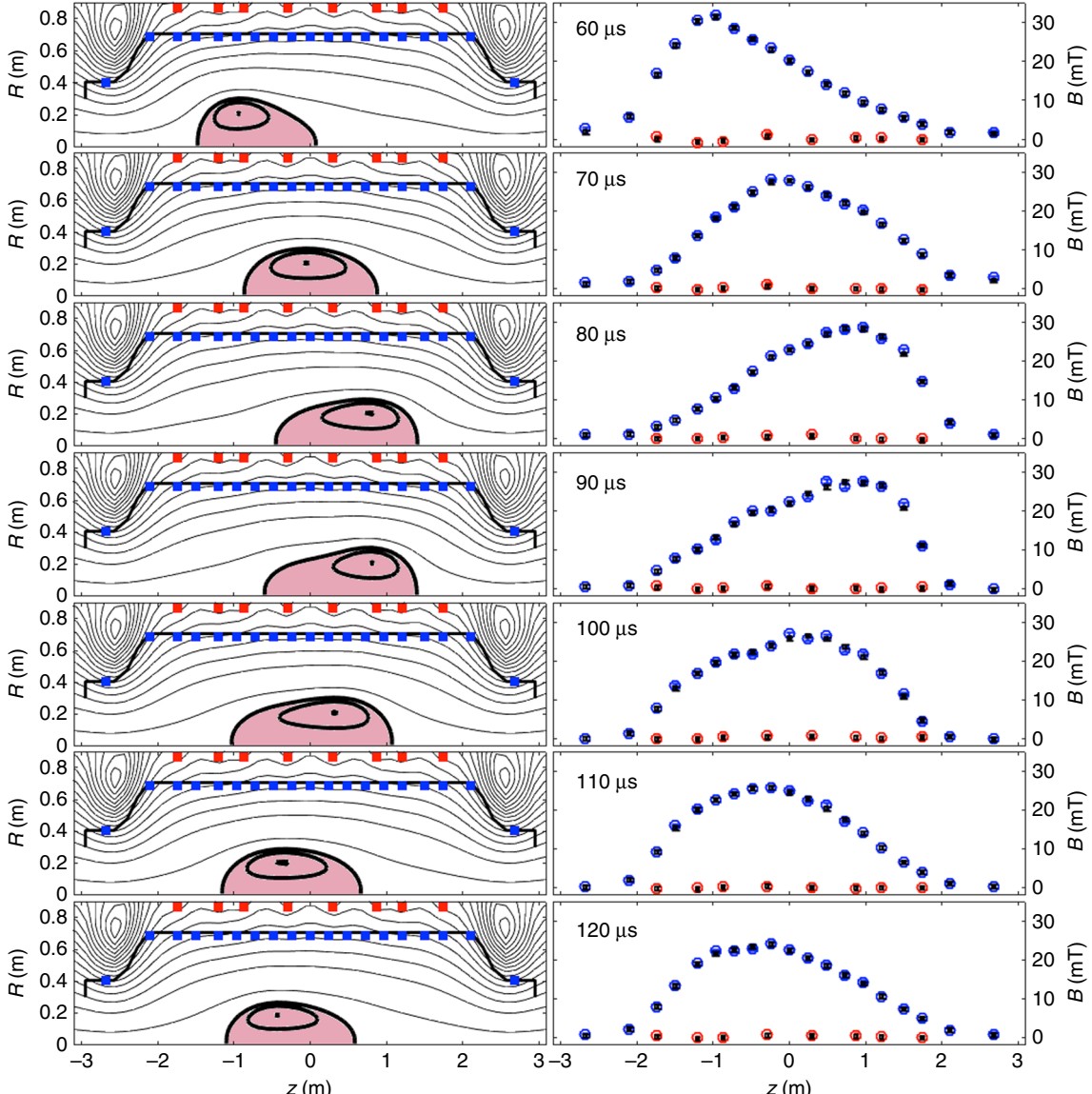

**Fig. 3** Inference of Alfvenic oscillations. Poloidal flux structure at the start of the C-2U shot #49040. Left panels: Contour map of the poloidal flux and its evolution in 10 µs intervals. External (red squares) and internal (blue squares) magnetic sensor locations are shown along with the vacuum vessel contour (black) intersecting the flux contours. Right panels: Magnetic field predictions (black) superimposed with the corresponding external (red) and internal (blue) probe measurements (after the magnetic field offset at $t = 0$ is subtracted). The forward prediction of the measurements is so accurate that differences with actual measurements are barely distinguishable. The frequency of the oscillations around the mid-plane is approximately 20 kHz. The flux-conserving effect of the vessel on this fast scale is evident from the absence of magnetic field change on the external probes

within one standard deviation, as illustrated in Fig. 5. So Newton's second law is recovered from the inference results. The algorithm, however, is not very accurate during the first 50 µs or so of the discharge (right after formation/merging) presumably because the smoothing prior used cannot adequately describe the abrupt profiles resulting from shock waves or violations of other prior assumptions.

Another test of relevance is to check whether the axial forces are proportional to some measure of plasma position $z$

$$\frac{\partial F_z}{\partial z} \cong \frac{F_z}{z} \qquad (3)$$

If Eq. (3) is valid for some axial range, a Hooke's constant can be defined. For a rigid plasma current distribution subjected to an

infinitesimal displacement, the Hooke's constant can be evaluated from the plasma current distribution and the externally applied flux $\psi_{ext}$ (from magnets and FC currents) as an integral extending over the plasma domain $\Omega$[21]

$$k_z = -\frac{\partial F_z}{\partial z} = 2\pi \int \int j_\phi \frac{\partial^2 \psi_{ext}}{\partial z^2} \mathrm{d}r\mathrm{d}z \qquad (4)$$

Note that when taking derivatives the flux created by the plasma does not change with $z$, as the plasma is considered a rigid object; only the external flux does change due to the relative motion.

A positive Hooke's constant corresponds with a magnetic configuration that is axially stable and vice versa. The evolution

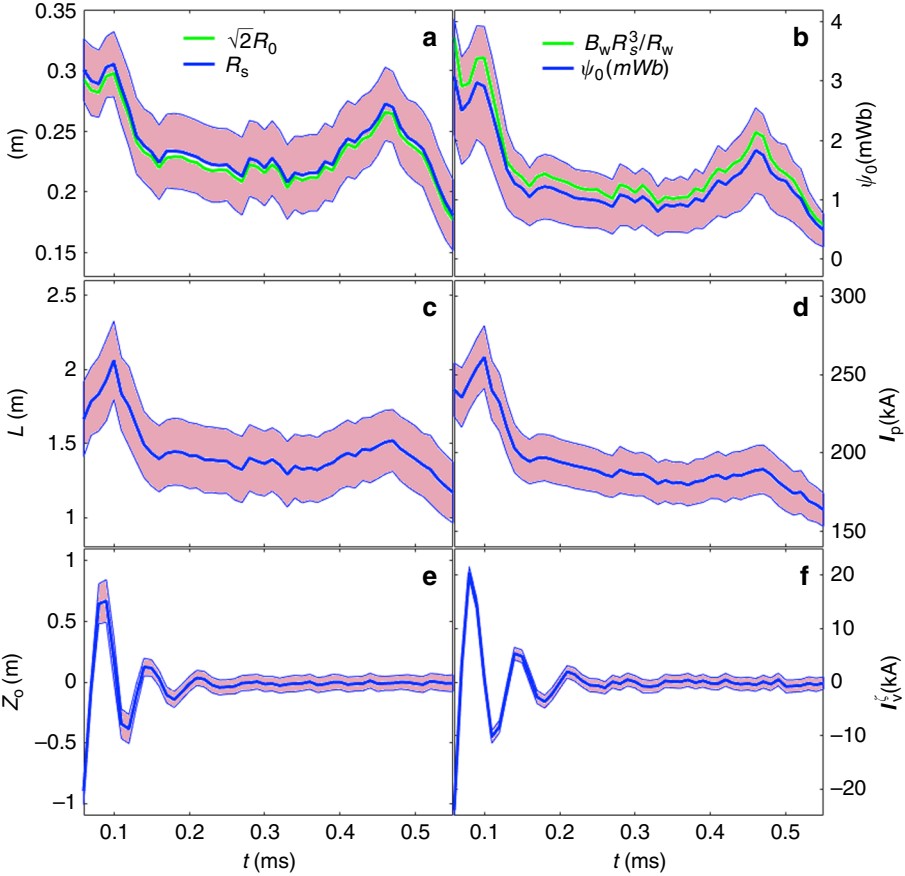

**Fig. 4** Inference of Alfvenic oscillations. Inferred plasma variables at the start of the C-2U shot #49040. Solid lines correspond with the expected values of the posterior distribution, and shaded areas are a measure of the uncertainty of the inferred variables corresponding to one standard deviation. **a** Separatrix radius $R_s$ (blue) and o-point radius $R_O$ times $\sqrt{2}$ (green). $R_s$ is found to be proportional to the $R_O$, in agreement with Eq. (7). **b** Trapped flux (blue) and the trapped flux approximation for a long FRC (green). **c** Separatrix length (distance between x-points). **d** Total plasma current. **e** o-point $z$ position. **f** Vessel current imbalance

on Hooke's constant and the axial force exerted on the plasma as a result of its axial displacement are shown in Fig. 5. Axial force and displacement are linearly dependent in a range of +/−1 m around the mid-plane, so plasma dynamics can be approximated by a linear partial differential equation in this range. This is interesting for the future plasma control goals, since control theory and practice are well established for linear systems[22].

The Hooke's constant is positive due to the axially stabilizing external field and therefore consistent with an axially stable magnetic configuration that reaches the mid-plane after a few oscillations, as observed. The inferred value of about 1000 N/m is in agreement with the results obtained using the Lamy Ridge code[23]. The inference method can also provide the axial stability properties of the magnetic configuration. This is an important information for plasma control of future devices, which will require to establish and sustain an axially unstable plasma in equilibrium around the mid-plane $z = 0$[24]. A method to determine the axial stability properties of the magnetic configuration will be therefore required.

**Comparison with plasma imaging.** High-speed imaging of visible plasma emission is an independent technique that can yield information about the plasma dimensions. In this study, qualitative agreement between visible light emission from intrinsic oxygen impurity ions and the dynamics of the inferred poloidal flux contours serve as additional validation of the proposed inference method. Photons emitted from the 3d→3p transition (at 650.0 nm) of $O^{4+}$ were measured using a filtered high-speed camera with a radial view of the plasma[25].

Emissivity of this spectral line was reconstructed (assuming axis symmetry) using the Simultaneous Algebraic Reconstruction Technique[26]. The core FRC electron temperature and density are more than sufficient to ionize the $O^{4+}$ charge state and populate higher charge states via electron impact excitation; therefore, minimal emission from this spectral line is found in the core. Instead, emission is peaked in the SOL where the electron temperature and density are lower and diffusive transport from boundary sources competes with ionization to higher charge states.

An example comparing the results of the magnetic inference method with the emissivity reconstruction for this spectral line is shown in Fig. 6. A relatively high-density plasma discharge (#48269) was chosen so that the emission measurement had good signal. Good agreement in the temporal dynamics of the reconstructed poloidal flux and emissivity is observed. This agreement provides further validation of the proposed inference method and is all the more encouraging since the two reconstructed quantities are derived from independent measurements (magnetics vs. photons) and analysis procedures.

The overall consistency of inferred results (Fig. 7) is also good, with the following highlights: (a) $R_s = \sqrt{2}R_0$ is really a very good approximation, within one sigma. (b) The long FRC trapped flux (Eq. (9)) $\psi_0 = B_w R_s^3 / R_w$ is also a very good approximation, being its overall magnitude in agreement with similar results obtained

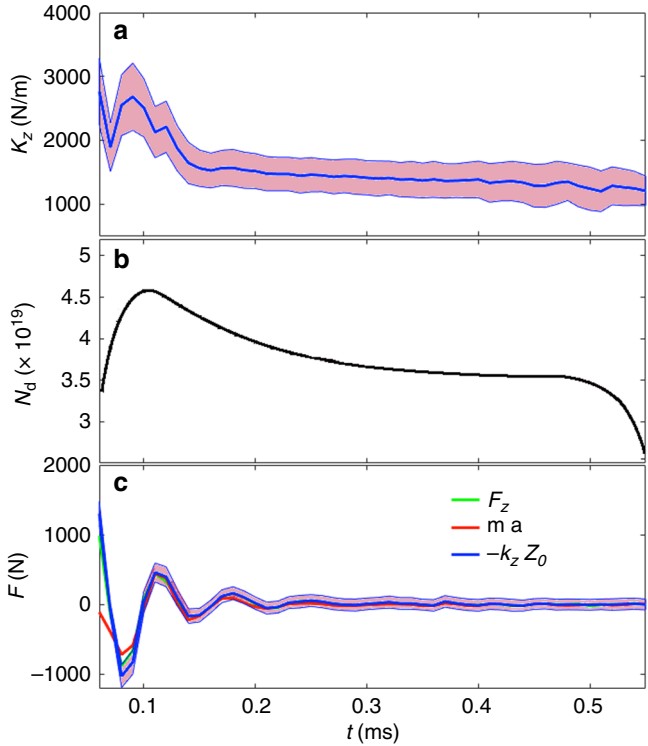

**Fig. 5** Inference of Alfvenic oscillations. **a** Hooke's constant as obtained from the inferred magnetic configuration. **b** Estimated particle inventory as stored in the C-2U database. **c** Electromagnetic force (solid green), plasma mass times acceleration (red) and Hooke's constant times axial plasma displacement (blue). The shaded area corresponds to one standard deviation

with hybrid and Grad–Shafranov equilibrium codes[27, 28]. (c) The magnitude of field reversal on axis is very significant and consistent with radial force balance (Eq. (10)) predictions. (d) The approximation (14) does not reproduce well the inferred FRC length. (e) There is a correlation between FRC length and plasma current, as expected from Eq. (11). However, the approximation (11) does not reproduce well the inferred results, partly because this approximation does not consider a current distribution flowing outside the separatrix. (f) Vessel current decays in about ~ 5 ms, comparable with the characteristic time over which the FRC is passively stabilized.

## Discussion

We have used the CT method to provide a direct inference of the internal FRC magnetic topology, both during steady state and fast Alfvenic transients. The viability of the approach has been verified in a number of ways, including comparisons with approximate results from a long FRC approximation, recovering of a force balance dynamic equation, and comparison with imaging of visible plasma emission.

All current sources have been modelled as GPs and inferred from external magnetic measurements using Bayesian analysis. Smoothing priors (for plasma current and vessel current distributions) and a flux-conserving prior derived from Lenz's law (for the magnet currents) have been used in the inference. From all the inferred current sources, FRC topology and dynamic properties have been obtained. This includes the separatrix geometry and the axial stability properties of the magnetic configuration, among others.

When GP priors are used, and linear relationships among current sources and measurements can be established, the CT solution involves non-iterative matrix operations and is then ideally suited for deterministic real-time applications. Because no equilibrium assumptions are used in this case, inference of plasma topology and dynamics up to Alfvenic frequencies then becomes possible. The FRC topology and dynamics have been determined during Alfvenic oscillations, with excellent self-consistency of results.

## Methods

**FRC approximations**. The inference results of experimental data presented have been compared with first-order approximations for FRC parameters, which are summarized below. These are valid for an elongated FRC inside a FC of constant radius $R_w$ with negligible field line curvature at the mid-plane, termed the long FRC approximation.

The radial pressure balance condition relates the magnetic field component in the axial direction right beneath the inner vessel walls at the o-point plane $B_w$ with the average kinetic pressure of the plasma:

$$P(r) \cong \frac{B_w^2 - B^2(r)}{2\mu_0}. \tag{5}$$

So the average kinetic pressure of the plasma at the o-point (where the magnetic pressure is null) must necessarily be equal to the magnetic pressure at the confinement vessel walls

$$P(0 - \text{point}) = \frac{B_w^2}{2\mu_0}. \tag{6}$$

When the plasma pressure is a flux function, and Eq. (5) is fulfilled, then the o-point radius is proportional to the separatrix radius 1.

$$R_0 \cong \frac{R_s}{\sqrt{2}}. \tag{7}$$

In addition to the former, the plasma is in axial force balance, and the maximum beta achievable by an ideal FRC surrounded by a perfect FC of constant radius $R_w$ is given by the Barnes' average $\beta$ condition[29],

$$\beta \cong 1 - 0.5\left(\frac{R_s}{R_w}\right)^2 \tag{8}$$

which depends solely on the ratio of separatrix radius $R_s$ to FC wall radius $R_w$.

When both axial and radial pressure balance are fulfilled, the flux at the o-point (trapped flux) is given by

$$\psi_0 \cong \frac{B_w R_s^3}{R_w}. \tag{9}$$

The plane perpendicular to the machine axis that contains the o-point is termed the o-point plane. The intersection of the o-point plane with the machine axis determines the point were the axial component of the magnetic field is minimum. From Eqs. (5) and (9), the magnitude of the field reversal $B_{ax}$ at this point is

$$B_{ax} \cong -\frac{R_s}{R_w}B_w \tag{10}$$

From Ampere's law and Eq. (10), the total plasma current in a very elongated FRC of length $L$ can be approximated by

$$I \cong \frac{(B_w - B_{ax})L}{\mu_0} \cong \frac{B_w}{\mu_0}\left(1 + \frac{R_s}{R_w}\right) \tag{11}$$

The plasma elongation is defined as

$$E = \frac{L}{2R_s}. \tag{12}$$

A common approximation to separatrix radius and length comes from the excluded flux radius axial profile, which can be derived directly from magnetic sensors[30], as explained below.

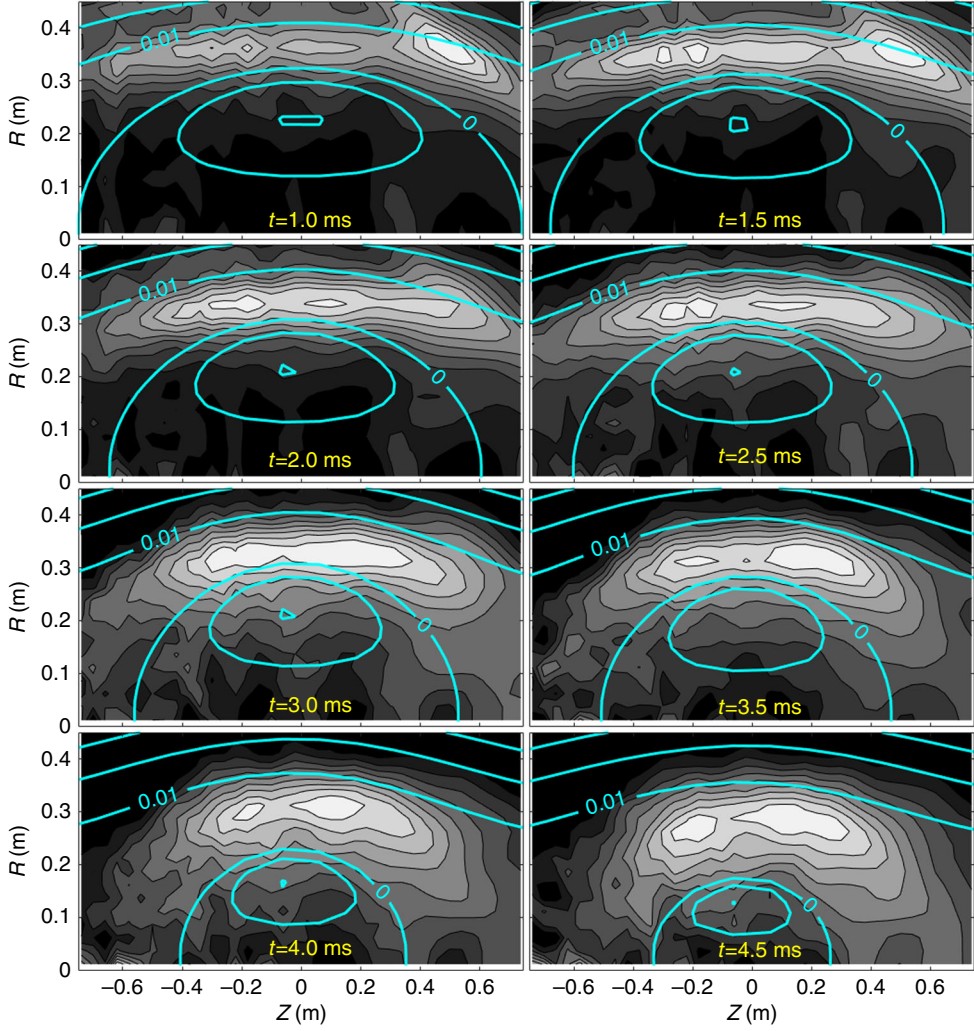

**Fig. 6** FRC poloidal flux contours. Flux contours (cyan) superimposed with surfaces of equal emissivity from the 3d→3p oxygen 4+ transition (grey scale). Maximum emissivity is shown in white, while no emissivity is represented by black. The flux contour levels corresponding with 0 Wb (separatrix) and 0.01 Wb are labelled to show how flux expands in the open field region as the discharge evolves. The region of peak emissivity approximately tracks the flux expansion

If $\psi^i$, $B_z^i$ are the flux and field profiles determined at positions $z^i$ along the internal wall of the vacuum vessel with radius $r = R_w$, the excluded flux radius profile is defined as

$$R_\psi^i = R_w \sqrt{1 - \frac{\psi^i}{\pi R_w^2 B_z^i}}. \tag{13}$$

The position of the plasma mid-plane can then be estimated from the position where the excluded flux radius has its maximum.

A first-order approximation for the FRC separatrix radius is to consider it equal to the excluded flux radius at the mid-plane. A first-order approximation for the FRC x-point position is taken as the point along the axis $Z_{2/3}$ where the excluded flux radius has fallen to 2/3 of its value at the mid-plane[28]. The FRC length is then approximated by

$$L = 2Z_{2/3}. \tag{14}$$

**Bayesian inference of GPs**. Bayesian inference is used in this paper to calculate the posterior distribution of currents given the magnetic measurements. The method, however, is generic enough to be used in a variety of related tomographic problems, which can be stated as follows.

Given a forward model $\mathbf{D} = H(X)$ relating a continuous variable $X(\mathbf{r})$ function of location $\mathbf{r} = (r_1, r_2, r_3)$ with some discrete set of measurements in the data vector $\mathbf{D}$, the objective is to obtain all the solutions for $X(\mathbf{r})$ that can explain the

measurements in $\mathbf{D}$. These are arranged into a probability distribution $p(X|\mathbf{D})$ termed the posterior. A likelihood probability distribution $p(\mathbf{D}|X)$ measures the misfit between the model predictions $H(X)$ and the measurements $\mathbf{D}$. The probability of the spatial variable $p(X)$ prior to taking any measurements is termed the prior probability distribution. According to Bayes theorem[14], the posterior can be obtained from the likelihood distribution and the prior as

$$p(X|\mathbf{D}) = \frac{p(\mathbf{D}|X)p(X)}{p(\mathbf{D})}. \tag{15}$$

The term in the denominator $p(\mathbf{D})$ is called the evidence (or marginal likelihood) and normalizes the volume of the posterior distribution to 1.

$$p(\mathbf{D}) = \int p(\mathbf{D}|X)p(X)\mathrm{d}X. \tag{16}$$

Given prior and likelihood, the most likely solution is the maximum a posteriori (MAP) estimate, the solution in the posterior with the highest probability.

In the particular case where the forward model is linear, the spatial variable $X(\mathbf{r})$ can always be discretized on a fine grid of dimension $k$, and a matrix $\mathbf{K} \in \mathbb{R}^{n \times k}$ can be used to relate the discretized variable $\mathbf{X} \in \mathbb{R}^k$ with a set of n measurements in $\mathbf{D} \in \mathbb{R}^n$

$$\mathbf{D} = \mathbf{KX} + \boldsymbol{\varepsilon} \tag{17}$$

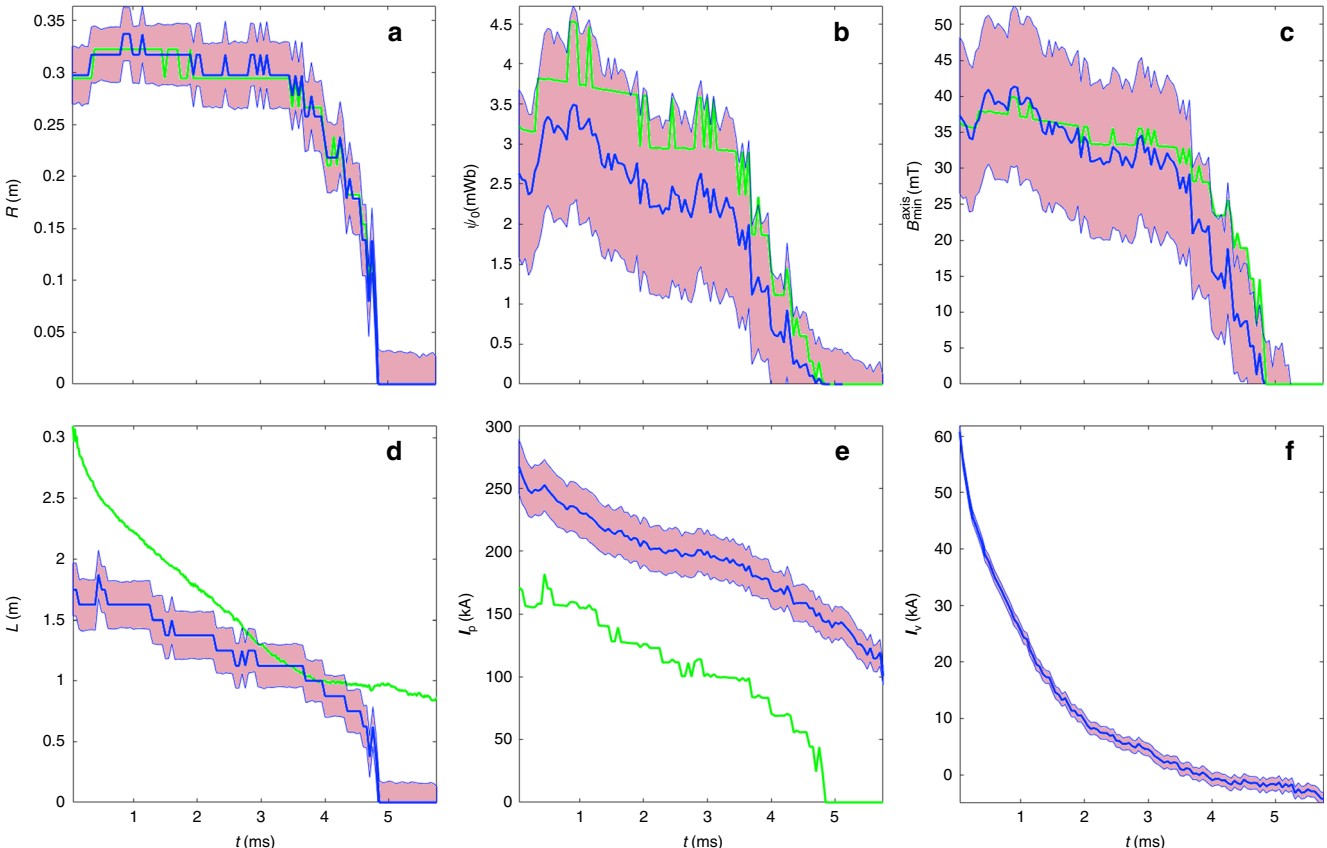

**Fig. 7** Inference of plasma magnetic related variables. Solid lines correspond with mean values of the posterior distribution, and shaded areas are a measure of the uncertainty of the inferred variables corresponding to one standard deviation for shot # 48269. **a** Separatrix radius $R_s$ (blue) and o-point radius $R_O$ times $\sqrt{2}$ (green). **b** Trapped flux (blue) and the trapped flux approximation for a long FRC (green). **c** Machine axis Bz at the o-point z position (blue) and comparison with the long FRC approximation (green). **d** Separatrix length (blue) compared with the approximation from excluded flux (green). **e** Total plasma current (blue) and comparison with the long FRC approximation (green). **f** Total vessel current

Assuming additive Gaussian measurement noise $\boldsymbol{\varepsilon} = N(0, \boldsymbol{\Sigma}_D)$ independent of $\mathbf{X}$, the likelihood function can be modelled by an $n$-dimensional Gaussian distribution

$$p(\mathbf{D}|\mathbf{X}) = \frac{1}{(2\pi)^{n/2}|\boldsymbol{\Sigma}_D|^{1/2}} \exp\left(-\frac{1}{2}(\mathbf{D} - \mathbf{K}\boldsymbol{X})^{\mathrm{T}}\boldsymbol{\Sigma}_D^{-1}(\mathbf{D} - \mathbf{K}\boldsymbol{X})\right) \quad (18)$$

where $\boldsymbol{\Sigma}_D \in \mathbb{R}^{n \times n}$ is the data covariance matrix.

The prior distribution can also be approximated by a multivariate probability distribution over $\overline{X}$

$$p(\mathbf{X}) = \frac{1}{(2\pi)^{k/2}|\boldsymbol{\Sigma}_X|^{1/2}} \exp\left(-\frac{1}{2}(\mathbf{X} - \boldsymbol{\mu}_X)^{\mathrm{T}}\boldsymbol{\Sigma}_X^{-1}(\mathbf{X} - \boldsymbol{\mu}_X)\right) \quad (19)$$

where $\boldsymbol{\Sigma}_X \in \mathbb{R}^{k \times k}$ is the prior covariance kernel and $\boldsymbol{\mu}_X \in \mathbb{R}^k$ is the prior mean. It is usually convenient (but by no means necessary) to consider a zero mean $\boldsymbol{\mu}_X = 0$ on the prior.

The posterior distribution can likewise be approximated by a $k$-dimensional Gaussian probability distribution.

$$p(\mathbf{X}|\mathbf{D}) = \frac{1}{(2\pi)^{k/2}|\boldsymbol{\Sigma}|^{1/2}} \exp\left(-\frac{1}{2}(\mathbf{X} - \boldsymbol{\mu})^{\mathrm{T}}\boldsymbol{\Sigma}^{-1}(\mathbf{X} - \boldsymbol{\mu})\right). \quad (20)$$

Since all probability distributions are Gaussian, the posterior distribution can be obtained analytically, since Gaussian distributions are transformed into Gaussian distributions through linear operations. The posterior mean (MAP estimate) and covariance are given in this case by[11]:

$$\boldsymbol{\Sigma} = \left(\mathbf{K}^{\mathrm{T}}\boldsymbol{\Sigma}_D^{-1}\mathbf{K} + \boldsymbol{\Sigma}_X^{-1}\right)^{-1}, \quad (21)$$

$$\boldsymbol{\mu} = \boldsymbol{\Sigma}\mathbf{K}^{\mathrm{T}}\boldsymbol{\Sigma}_D^{-1}\mathbf{D}. \quad (22)$$

As the dimension $k$ of the multivariate normal distribution is made increasingly large, the multivariate normal distribution approaches a continuous distribution, and at this limit a GP is obtained[15]. In our case, the vector $\mathbf{X}$ becomes a continuous function $X(\mathbf{r})$ of the spatial location. All possible solutions for $X(\mathbf{r})$ can then be thought of as being generated by a stochastic process, described by the corresponding GP.

For a large number of situations in plasma physics, transport processes will work in the direction to reduce the spatial gradients of $X(\mathbf{r})$. In other words, our prior belief about $X(\mathbf{r})$ is that it must be a smooth function of $\mathbf{r}$. The prior covariance kernel required for the inference can then be parameterized using the Squared exponential (SE) function, which is one of many options available[15] to model the spatial correlations between the values of a smooth profile variable at two points $\mathbf{r}$ and $\mathbf{r}'$:

$$\boldsymbol{\Sigma}_X(\mathbf{r}, \mathbf{r}') = \sigma^2 \exp\left(-\frac{1}{2}(\mathbf{r} - \mathbf{r}')^{\mathrm{T}}\boldsymbol{\Lambda}^{-1}(\mathbf{r} - \mathbf{r}')\right) \quad (23)$$

with $\boldsymbol{\Lambda} = \mathrm{diag}(\lambda_1^2, \lambda_2^2, \lambda_3^2)$.

In the Bayesian context, $\sigma$ and $\lambda_i$ are termed the prior hyper-parameters. The standard deviation $\sigma$ controls the spread of values of $X$. The scale length $\lambda_i$ determines how quickly the plasma variable can change with the coordinate $r_i$. A large length scale will give a large covariance between the values of the variable $X$ at different $r_i$ coordinates, so the prior probability (Eq. (19)) for large differences between the values of the plasma variable $X$ at neighbouring positions $r_i$, $r'_i$ will be low. In other words, if the plasma profiles are smooth, the corresponding scale lengths will be large, and vice versa.

The SE kernel is by no means the only choice for a prior covariance kernel. A good review of GPs and the most common covariance kernels used can be found in ref. [15]. In general, the prior covariance kernel will have a set of hyper-parameters, arranged in a vector $\boldsymbol{\theta}$ for simplicity.

Determination of the prior hyper-parameters can be considered as a continuous model selection problem, where the more likely hyper-parameters are obtained directly from the data[31].

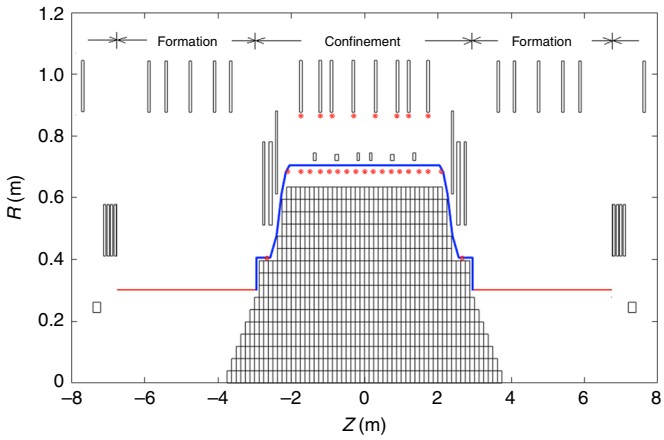

**Fig. 8** C-2U magnetostatic model. Toroidal plasma current is modelled using 734 discrete plasma current elements modelled as block coils (back grid inside the vessel contour in blue). The vessel is modelled using 31 flat block coil elements (not shown). Insulating quartz tubes (formation sections) are shown in red. All the magnets in confinement and formation sections are considered. Pulsed powered fast switching coils for plasma formation and acceleration (not shown) are located right outside the quartz tube. Magnetic probes inside and outside the confinement vessel are shown as red circles

The posterior for the hyper-parameters is $p(\boldsymbol{\theta}|\mathbf{D})$, which from Bayes theorem is

$$p(\boldsymbol{\theta}|\mathbf{D}) = \frac{p(\mathbf{D}|\boldsymbol{\theta})p(\boldsymbol{\theta})}{p(\mathbf{D})} \tag{24}$$

where $p(\mathbf{D}|\boldsymbol{\theta})$ is the likelihood and $p(\boldsymbol{\theta})$ is the hyper-prior (prior for the hyper-parameters).

In Bayesian model selection, the optimum set of hyper-parameters $\boldsymbol{\theta}_{opt}$ is selected to maximize this probability.

$$\boldsymbol{\theta}_{opt} = \arg_{\boldsymbol{\theta}} \max(p(\boldsymbol{\theta}|\mathbf{D})) \tag{25}$$

The prior over the hyper-parameters $p(\boldsymbol{\theta})$ in Eq. (24) is usually taken to be flat, since there is no indication of what are the best hyper-parameters before seeing the data. In this case, the optimal set of hyper-parameters that maximizes likelihood of the data with respect to the hyper-parameters is

$$\boldsymbol{\theta}_{opt} = \arg_{\boldsymbol{\theta}} \max(p(\boldsymbol{\theta}|\mathbf{D})) = \arg_{\boldsymbol{\theta}} \max(p(\mathbf{D}|\boldsymbol{\theta})) \tag{26}$$

Given a set of hyper-parameters $\boldsymbol{\theta}$, there is an infinite class of plasma profiles $X$ (**r**) that can be generated by the corresponding prior covariance $p(X|\boldsymbol{\theta})$ through the corresponding GP. The quality of the data fit must be evaluated not just for one particular solution but for all the solutions that can be obtained for a given set of hyper-parameters. The likelihood should be integrated out (marginalized) with respect to all these possible profiles generated by a single set of hyper-parameters, so it becomes a marginal likelihood.

$$\boldsymbol{\theta}_{opt} = \arg_{\boldsymbol{\theta}} \max(p(\mathbf{D}|\boldsymbol{\theta})) = \arg_{\boldsymbol{\theta}} \max\left(\int p(\mathbf{D}|X,\boldsymbol{\theta})p(X|\boldsymbol{\theta})\mathrm{d}X\right) \tag{27}$$

In the particular case at hand where $p(X)$ is a GP, the likelihood is normal and the model linear, the marginal likelihood can be calculated analytically. The expression for its logarithm is[16]

$$L = -\frac{1}{2}\log\left|\mathbf{K}\Sigma_X\mathbf{K}^T + \Sigma_D\right| - \frac{1}{2}\mathbf{D}^T\left(\mathbf{K}\Sigma_X\mathbf{K}^T + \Sigma_D\right)^{-1}\mathbf{D} - \frac{n}{2}\log(2\pi). \tag{28}$$

For any given prior kernel, the maximum of the expression (28) with respect to the hyper-parameters gives the optimal set of hyper-parameters that explain **D**.

**Inference model for the C-2U device**. The C-2U magnetic model used for the analysis comprises a total of 42 magnets, 31 vacuum vessel (passive) segments and a current distribution made of 734 discrete plasma current elements modelled as block coils (Fig. 8). Of special relevance are 8 equilibrium (EQ) magnets in the confinement vessel and 6 FC magnets, which can be used as passive FCs or be connected to power supplies.

The magnetic measurement system on C-2U[32, 33] comprises a set of 19 magnetic pick-up probes placed inside the confining vessel and 8 external pick-ups located right underneath the 8 EQ magnets (Fig. 8). There are also Rogowski-based current measurements for all the FC magnets currents $\mathbf{I}_{EQ}$ and also for some of the EQ magnets currents $\mathbf{I}_{EQ}$. For the rest of the magnets, only the set point used for its control is known.

The inference problem at hand requires finding the most likely solution for the elements of the total plasma current distribution arranged in a vector $\mathbf{I}_P$, along with the most likely solution for current induced in the confining vessel $\mathbf{I}_V$ and all the magnets $\mathbf{I}_M$. A diagram illustrating the magnet location and grid used for the current distribution is shown in Fig. 8.

All the currents to be inferred are arranged into a single current vector

$$\mathbf{I} = \{\mathbf{I}_P, \mathbf{I}_V, \mathbf{I}_M\}. \tag{29}$$

All current sources are modelled as GPs as described earlier. The information available to perform the inference comes from (i) set points for all $\mathbf{I}_M$, (ii) current measurements for $\mathbf{I}_{FC}$ and a few $\mathbf{I}_{EQ}$, (iii) measurements of magnetic field at several locations outside the plasma region, both inside and outside the confining vessel, (iv) null boundary conditions for plasma current distribution, and (v) null boundary conditions for the flux change underneath the equilibrium magnets $\frac{\partial \psi}{\partial t} \cong 0$, which behave as perfect FCs on the timescale of the discharge.

The boundary conditions (iv) and (v) are built directly into the prior, to obtain solutions where the plasma current distribution drops to zero at the domain, and the flux is conserved at the magnet locations (flux-conserving prior).

From the inferred currents in **I** is then straightforward to calculate the poloidal flux and magnetic field components on the domain grid using the matrix representations $\mathbf{M}, \mathbf{G}_R, \mathbf{G}_Z$ of the Biot–Savart operator[34]:

$$\begin{aligned} \boldsymbol{\psi} &= \mathbf{M}\,\mathbf{I}; \\ \mathbf{B}_R &= \mathbf{G}_R\mathbf{I}; \\ \mathbf{B}_Z &= \mathbf{G}_Z\mathbf{I}. \end{aligned} \tag{30}$$

Main plasma shape and position variables of interest for control such as x-point, o-point and separatrix radius can then be obtained directly by searching for nulls on the magnetic field and flux along the axis and mid-plane. Low-order moments of the plasma current distribution of interest for control such as total plasma current or the axial position of current centroid can likewise be obtained from linear operations.

$$\begin{aligned} I_P &= \mathrm{sum}(\mathbf{I}_P); \\ z_0 I_P &= \mathbf{z}^T\mathbf{I}_P \quad . \end{aligned} \tag{31}$$

**Data availability**. All relevant data supporting the findings of this study are available from the authors on request.

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

## Acknowledgements

This work has been performed thanks to the continued financial support from TAE's investors. The authors are very grateful to R. Mendoza for database support. The authors would like to acknowledge helpful discussions with M. Onofri, R. Smith, L. Steinhauer, E. Trask, S. Putvinski and P. Yushmanov.

## Author contributions

J.A.R. developed the Bayesian inference framework, performed the data analysis and wrote the manuscript. S.A.D. provided equilibrium simulation results used for validation of the inference method at its early stage. E.G. provided the SART-based tomographic inversion for plasma emissivity. T.R. provided magnetic probes and magnet current measurements. Y.M. made the simulations of formation/merging.

## Additional information

**Competing interests:** The authors declare no competing financial interests.

