## [Peer Review File · Nature Communications]

Reviewers' comments:

Reviewer #1 (Remarks to the Author):

Overall, I thought the work itself was interesting and useful to those in the field. I recommend it be published, with some editing to improve the flow and impact of the work.

I did feel that the organization was confusing, requiring a good deal of jumping around. Intro, methods, results and then discussion would make more sense to the majority of readers. The exception would be readers already well versed in FRC physics and Bayesian inference methods. I think the work is worthwhile because most plasma physicists who would be in the position to use these results are NOT versed in the statistical technique and haven't thought through needs of control physics. If the order can't be changed, it still works, perhaps the methods sections could be referenced in the earlier text, something like, 'FRC model is described in section 4a.

Second, I feel that more context would be warranted, with detail added in the intro and more importantly the discussion section. Has this technique been implemented before in plasma systems? Why not? What has enabled the use of the technique now? What is the computational requirement and what are the limits for control in terms of frequency? What is the impact of this technique? What could you not do without it? How about extensibility? Could it work for non-linear problems? Are there practical limits? For example, how sensitive is the technique to certain assumptions, like axisymmetry? A specific question is that of starting conditions, for Fig. 7, does the technique track formation? Are there concerns with tracking a complex start-up? How long does it take for the algorithm to 'lock-in' to a solution? Does uncertainty in the solution take some time to level-out to a steady or normal value? I'm wondering since it relies on prior information. Does it ever not work?

To sum up, the main claim is that this technique is ideally suited for real-time control, so I'd like a more quantified claim on its suitability. The work is already done, just need to call out the metrics...speed, determinancy and accuracy come to mind.

Detailed comments/questions:

line 114: Where is the correlation shown?

line 115: forward referencing makes it hard to read

line 140: better to avoid 'our' personal pronouns

Fig. 3: black field lines do not appear to be accurate, they are not updated with the plasmoid motion. If this is a cartoon, that is fine...or maybe it really doesn't change, but that doesn't seem to square with the measurements at the locations shown. I would think the entire field structure would be found based on the measurements.

Fig. 4: If the method gives a measure of error or uncertainty, would that not be a better choice to show than a simple standard deviation?

line 197: can 'good approximation' be quantified?

line 244: reference for 'Barnes average beta condition'

line 270-350: The section seems a little 'separate' some language at the beginning and end of the section would improve the flow, giving the link to the preceding and following sections.

Similar comment to the inference model section, line 351-383...here again, something to wrap up the section, for instance highlighting how the reconstruction then is used.

line 375,

line 382...reference for the Biot-Savart operator?

Reviewer #2 (Remarks to the Author):

Overall a very well written publication, providing first direct inference on the internal FRC magnetic topology, and also showing, another first, that the magnetic topology can be inferred during fast transient phenomena, which would not be possible with standard approaches based on equilibrium assumptions. The viability of the approach has been verified in a number of ways, including comparisons with approximate results from a long FRC approximation, recovering of a force balance dynamic equation, and comparison with independently measured plasma emission distribution. The paper is methodologically very sound and represents a novel contribution to the understanding of FRC magnetic topology and dynamics. Please find some minor comments below.

Line 14-15:

“a physics model in the prior.”

I would advice not claiming there is a physics model in the prior, when the prior is a Gaussian process.

Line 20

“becomes then possible”, I believe “then becomes possible” is better grammatically.

Page 4, equation 4

Write a double integral sign and the integration limits.

I would also like to see an explanation of why the vacuum flux rather than the actual flux is used in the expression.

Figure 4.

I can't find any explanation of the inclusion of the factor $\sqrt{2}$. A forward reference to equation 7 would be helpful.

Figure 5

Can the inital discrepancy near $t=0$ between plasma mass times accelaration (red) and the Lorentz force (green) be explained?

Line 186 and Line 195

The wording “excellent consistency” and “closely tracks” I think is too strong. In figure 6 the emissivity certainly shows good agreement with the region covered by the open field lines, but I think the argument would be good enough if the agreement was described as “good agreement” or something similar, since otherwise the expectation would be that the agreement is perfect. For example, a quite large variation in the separatrix radius would still show similarly good agreement with the emission (at for example $t=2.0\text{ms}$, $t=2.5\text{ms}$ or $t=4.0\text{ms}$).

Line 299

The posterior distribution can be likewise be approximated. Take away the first “be”.

Line 302

I would skip “always”, it is not needed.

Line 313

“our prior believe”, should be “our prior belief”

Line 315

I don't think it is appropriate to say the SE prior is “ideally suited” to model spatial correlations. There

are a number of other GPs that model spatial correlations in ways suited for different problem domains.

Line 335

"in is usually", take away "in"

Line 337

"maximizes likelihood of the data", should probably be "that maximizes the likelihood of the data"

Chapter 4 enters in the publication as an appendix and should possibly be renamed as that, but that is up to the authors.

Answer to referee 1

Generic comments

- Overall, I thought the work itself was interesting and useful to those in the field. I recommend it be published, with some editing to improve the flow and impact of the work.

We are glad the paper was found interesting, thanks for taking the time to review it.

- I did feel that the organization was confusing, requiring a good deal of jumping around. Intro, methods, results and then discussion would make more sense to the majority of readers. The exception would be readers already well versed in FRC physics and Bayesian inference methods. I think the work is worthwhile because most plasma physicists who would be in the position to use these results are NOT versed in the statistical technique and haven't thought through needs of control physics. If the order can't be changed, it still works, perhaps the methods sections could be referenced in the earlier text, something like, 'FRC model is described in section 4a.

We agree with your comments regarding paper organization.

However this was not our choice, but an editorial policy. Following your suggestions we have added a paragraph regarding paper organization. :

The paper starts directly with the inference results (section 2), followed by a discussion (section 3). An explanation of some common formulae used in FRC research along with the Bayesian inference method can be found at the end of the paper (section 4).

- Second, I feel that more context would be warranted, with detail added in the intro and more importantly the discussion section. Has this technique been implemented before in plasma systems? Why not? What has enabled the use of the technique now?

Context has been provided. Lines 54-59 provide references [11], [12], [16] where similar methods have been used in Tokamaks. We don't know why was this not applied before for FRC inference. A possible explanation is that an FRC with lifetime exceeding the violent formation merging process is required. This may not have been available prior to TAE's beam driven FRC technology.

- What is the computational requirement and what are the limits for control in terms of frequency?

The following sentence has been added in the introduction

(sampling frequency in the range 10-100 kHz)

- What is the impact of this technique? What could you not do without it?

There is not much we can do to control the plasma without knowing where the plasma is. Introduction states the need for this (lines 38-41), and some available techniques like excluded flux radius (lines 41-43). Rather than saying what we can do without it, we have focused on what we can do with it and why is ideally suited for the problem at hand (lines 54-74)

- How about extensibility? Could it work for non-linear problems? GP inference can be used for non-linear problems, but in this case the inference method requires McMM or variational methods, which make it impractical for real time applications.
- What are there practical limits? For example, how sensitive is the technique to certain assumptions, like axisymmetry? A specific question is that of starting conditions, for Fig. 7, does the technique track formation? Are there concerns with tracking a complex start-up? How long does it take for the algorithm to 'lock-in' to a solution? Does uncertainty in the solution take some time to level-out to a steady or normal value? I'm wondering since it relies on prior information. Does it ever not work?

There is no lock-in period with this technique, but results tend to have lower uncertainty past the formation merging process, roughly after 50us or so. The algorithm is not very accurate during the first 50us or so of the discharge, right after formation, presumably because the SE prior can not adequately describe abrupt profiles resulting from shock waves, or violations of other prior assumptions, like axisymmetry. This can already be appreciated in fig.5 . A discrepancy near $t=0$ is observed between plasma mass times acceleration (red) and the Lorentz force (green).

- To sum up, the main claim is that this technique is ideally suited for real-time control, so I'd like a more quantified claim on its suitability. The work is already done, just need to call out the metrics...speed, determinancy and accuracy come to mind.

For the purpose of position control (the most demanding) the algorithm has already been tested. The following sentence has been added:

A version of the algorithm for C-2W device has already been implemented in a field-programmable gate array (FPGA) and verified to run under 10us.

The method is deterministic as the inference requires a matrix multiplications (lines 63-64).

Regarding accuracy the model assumes Gaussian distributions, so the standard deviation for all the inferred variables in in fact the uncertainty measure.

Detailed comments/questions:

- line 114: Where is the correlation shown?
Comparison between plasma position (bottom left) and current imbalance (bottom right) of figure 4. They look identical in shape. The Figure 4 is referred earlier, lines 105-107.
- line 115: forward referencing makes to hard to read .
This is a result of the editorial requirements of placing the methods section at the end of the paper.
- line 140: better to avoid 'our' personal pronouns.
Substitued by “the”
- Fig. 3: black field lines do not appear to be accurate, they are not updated with the plasmoid motion. If this is a cartoon, that is fine...or maybe it really doesn't change, but that doesn't seem to square with the measurements at the locations shown. I would think the entire field structure would be found based on the measurements.
The black lines represent flux contours, which do not change in the proximity of the vessel on the time scale shown. This is the result of the flux conserving effect of the vessel. Flux lines, however, do compress as result of the plasma displacements, resulting in a higher magnetic field, but we acknowledge this could not be noticed in the figure provided. Figure 3 has been modified by adding more flux contours to make this effect more noticeable.
- Fig. 4: If the method gives a measure of error or uncertainty, would that not be a better choice to show than a simple standard deviation?
Actually, since the model assumes Gaussian distributions, the measure of error given by the method is in fact the standard deviation.
- line 197: can 'good approximation' be quantified?
The following has been added, “within one sigma”.
- line 244: reference for 'Barnes average beta condition'

The reference containing the derivation has been added

W. T. Armstrong et al (1981). Field reversed experiments (FRX) on compact toroids. Physics of Fluids 24, 2068.

- line 270-350: The section seems a little 'separate' some language at the beginning and end of the section would improve the flow, giving the link to the preceding and following sections.

The following sentence has been added:

Bayesian inference is used in this paper to calculate the posterior distribution of currents given the magnetic measurements. The method, however, is generic enough to be used in a variety of related tomographic problems, which can be stated as follows.

- Similar comment to the inference model section, line 351-383...here again, something to wrap up the section, for instance highlighting how the reconstruction then is used.

The following sentence has been added:

Main plasma shape and position variables of interest for control such as x-point, o-point and separatrix radius can then be obtained directly by searching for nulls on the magnetic field and flux along the axis and mid-plane. Low order moments of the plasma current distribution of interest for control such as total plasma current or the axial position of current centroid can likewise be obtained from linear operations.

- line 382...reference for the Biot-Savart operator?

The following reference has been added:

J.T.Conway. (2006). Trigonometric Integrals for the Magnetic Field of the Coil of Rectangular Cross Section. IEEE Transactions on Magnetics, Vol 42 , N5

Answer to referee 2

General comments

Overall a very well written publication, providing first direct inference on the internal FRC magnetic topology, and also showing, another first, that the magnetic topology can be inferred during fast transient phenomena, which would not be possible with standard approaches based on equilibrium assumptions. The viability of the approach has been verified in a number of ways, including comparisons with approximate results from a long FRC approximation, recovering of a force balance dynamic equation, and comparison with independently measured plasma emission distribution. The paper is methodologically very sound and represents a novel contribution to the understanding of FRC magnetic topology and dynamics. Please find some minor comments below.

We are glad the paper was found interesting, thanks for taking the time to review it.

Specific comments

- Line 14-15:
“a physics model in the prior.”
I would advice not claiming there is a physics model in the prior, when the prior is a Gaussian process.
The GP prior does not exclude the use of physics models. A flux conserving prior for instance, is used in this paper. Flux conservation is a result of Lenz’s law. This is described in the last section (lines 375 to 380), but not emphasized enough. The following text has been added.
The boundary conditions iv) and v) are built directly into the prior, to obtain solutions where the plasma current distribution drops to zero at the domain, and the flux is conserved at the magnet locations.
- Line 20
“becomes then possible”, I believe “then becomes possible” is better grammatically.
This has been corrected.
- Page 4, equation 4
Write a double integral sign and the integration limits.
I would also like to see an explanation of why the vacuum flux rather than the actual flux is used in the expression.

The Hooke's constant assumes a rigid current distribution profile subjected to an infinitesimal displacement along z . When taking derivatives the flux created by the plasma does not change with z , as the plasma is considered a rigid object, only the vacuum flux does change due to the relative motion.

The integral has been modified, and a sign error has been corrected.

- Figure 4.

I can't find any explanation of the inclusion of the factor $\sqrt{2}$. A forward reference to equation 7 would be helpful.

The following text has been added to the caption:

R_s is found to be proportional to the R_0 , in agreement with equation 7

- Figure 5

Can the initial discrepancy near $t=0$ between plasma mass times acceleration (red) and the Lorentz force (green) be explained?

The algorithm is not very accurate during the first 50 μ s or so of the discharge, right after formation, presumably because the SE prior can not adequately describe abrupt profiles resulting from shock waves, or violations of other prior assumptions.

- Line 186 and Line 195

The wording "excellent consistency" and "closely tracks" I think is too strong. In figure 6 the emissivity certainly shows good agreement with the region covered by the open field lines, but I think the argument would be good enough if the agreement was described as "good agreement" or something similar, since otherwise the expectation would be that the agreement is perfect. For example, a quite large variation in the separatrix radius would still show similarly good agreement with the emission (at for example $t=2.0$ ms, $t=2.5$ ms or $t=4.0$ ms).

This has been corrected in main text and caption.

- Line 299

The posterior distribution can be likewise be approximated. Take away the first "be".

Corrected

- Line 302

I would skip "always", it is not needed.

Corrected

- Line 313

"our prior believe", should be "our prior belief"

Corrected

- Line 315
I don't think it is appropriate to say the SE prior is "ideally suited" to model spatial correlations. There are a number of other GPs that model spatial correlations in ways suited for different problem domains.
Ideally suited has been substituted by "one of the many options available"
- Line 335
"in is usually", take away "in"
A reference to equation 24 was missing.
The text now reads "The prior over the hyper-parameters $p(\theta)$ in (24) is usually..."
- Line 337
"maximizes likelihood of the data", should probably be "that maximizes the likelihood of the data"
Corrected
- Chapter 4 enters in the publication as an appendix and should possibly be renamed as that, but that is up to the authors.
An editorial requirement is that this section should be named "methods", so it is really not our choice.

REVIEWERS' COMMENTS:

Reviewer #1 (Remarks to the Author):

My thanks to the authors for their thorough and thoughtful responses to my review and questions. I think the enacted changes adequately address the content and style concerns that I brought up. The paper is novel and of interest to the wider community, is well-written and supports its claims. As such, I recommend the work to be published.

Reviewer #2 (Remarks to the Author):

I am happy with the responses and changes to the document. My intention with the comment on Page 4, equation 4 and the comment about figure 5 was to include those explanations in the text rather than just explaining to me. Both explanations were satisfactorily though, and I leave it to the authors whether to include those.